# Structure-based design of a dual-warhead covalent inhibitor of FGFR4

Xiaojuan Chen[1,5], Huiliang Li[2,5], Qianmeng Lin[1], Shuyan Dai[1], Sitong Yue[1], Lingzhi Qu[1], Maoyu Li[1], Ming Guo[1], Hudie Wei [1], Jun Li[1], Longying Jiang [1,3✉], Guangyu Xu [2✉] & Yongheng Chen [1,4✉]

The fibroblast growth factor 19 (FGF19)/fibroblast growth factor receptor 4 (FGFR4) signaling pathways play critical roles in a variety of cancers, such as hepatocellular carcinoma (HCC). FGFR4 is recognized as a promising target to treat HCC. Currently, all FGFR covalent inhibitors target one of the two cysteines (Cys477 and Cys552). Here, we designed and synthesized a dual-warhead covalent FGFR4 inhibitor, **CXF-009**, targeting Cys477 and Cys552 of FGFR4. We report the cocrystal structure of FGFR4 with **CXF-009**, which exhibits a dual-warhead covalent binding mode. **CXF-009** exhibited stronger selectivity for FGFR4 than FGFR1-3 and other kinases. CXF-009 can also potently inhibit the single cystine mutants, FGFR4(C477A) and FGFR4(C552A), of FGFR4. In summary, our study provides a dual-warhead covalent FGFR4 inhibitor that can covalently target two cysteines of FGFR4. **CXF-009**, to our knowledge, is the first reported inhibitor that forms dual-warhead covalent bonds with two cysteine residues in FGFR4. **CXF-009** also has the potential to overcome drug induced resistant FGFR4 mutations and might serve as a lead compound for future anticancer drug discovery.

[1] Department of Oncology, NHC Key Laboratory of Cancer Proteomics, State Local Joint Engineering Laboratory for Anticancer Drugs, Xiangya Hospital, Central South University, Changsha, Hunan, China. [2] Key Laboratory of Chemical Biology and Traditional Chinese Medicine, Ministry of Educational of China, Key Laboratory of the Assembly and Application of Organic Functional Molecules of Hunan Province, College of Chemistry and Chemical Engineering, Hunan Normal University, Changsha, Hunan, China. [3] Department of Pathology, Xiangya Hospital, Central South University, Changsha, Hunan, China. [4] National Clinical Research Center for Geriatric Disorders, Xiangya Hospital, Central South University, Changsha, Hunan, China. [5]These authors contributed equally: Xiaojuan Chen, Huiliang Li. ✉email: longyingj1024@163.com; gyxu@hunnu.edu.cn; yonghenc@163.com

The family of fibroblast growth factor receptors (FGFRs) is comprised of four members (FGFR1-4)[1]. FGFRs play important roles in many physiological processes such as cell proliferation, differentiation, and tissue maintenance and repair through the activation by their ligands, fibroblast growth factors (FGFs)[2]. Upon binding to FGFs via the extracellular domain, FGFRs dimerize and trigger protein phosphorylation to activate the Ras-mitogen-activated protein kinase (MAPK) pathway and protein kinase B (AKT) pathways[3]. Dysregulation of the FGF/FGFR signaling axis is associated with carcinogenesis, tumor progression, and resistance to anticancer therapy in many tumor types[4]. FGFR aberrations, including gene amplification, mutations, and rearrangements, were found in 7.1% of cancers[5]. The cancers most frequently affected by FGFR aberrations were urothelial cancer (32% of FGFR abnormalities); breast cancer (18%); endometrial cancer (~13%); squamous lung cancer (~13%), and ovarian cancer (~9%)[5].

FGFR inhibition has been recognized as an important therapeutic option for the treatment of multiple tumor types. Erdafitinib, the first FDA-approved FGFR inhibitor, is indicated for patients with locally advanced or metastatic uroepithelial cancer that progresses during or after platinum-containing chemotherapy with an FGFR3 or FGFR2 mutation[6]. Pemigatinib, another FGFR inhibitor, was approved by the FDA in April 2020 for the treatment of patients with unresectable locally advanced or metastatic cholangiocarcinoma (CCA) with FGFR2 rearrangement or fusion[7,8]. Infigratinib (BGJ398), a pan-FGFR inhibitor, is approved to treat locally advanced or metastatic CCA in adults with an FGFR2 rearrangement[9,10]. Other pan-FGFR inhibitors, PRN1371 and Futibatinib (TAS-120), have been explored in clinical trials for the treatment of FGFR-driven cancers[11,12]. These inhibitors were designed to form a covalent bond with the cysteine 477, conserved among all FGFR paralogs, located in the P-loop of the FGFR ATP-binding pocket[13]. The side effects of these inhibitors, such as hyperphosphatemia owing to off-target effects on other kinases with the analogous p-loop cysteine (Fig. 1a), restrict their further development[14].

FGFR4 is mainly expressed in liver, lung, and breast tissues and specifically binds to the endocrine ligand FGF19[15]. Aberrant FGF19/FGFR4 signaling pathways (i.e., amplification, mutation, rearrangements, or overexpression) have been detected in a variety of cancers, such as hepatocellular carcinoma (HCC), breast cancer, prostate cancer, and pancreatic cancer[16].

Importantly, FGFR4 has the least homology among the FGFR family, enabling the design of FGFR4-specific inhibitors[17]. Therefore, targeting FGFR4 has been known as a promising therapeutic strategy for the development of small-molecule inhibitors for the treatment of cancers with aberrant FGFR4 activation[18].

FGFR4 has two cysteine residues, Cys477 and Cys552, located near the binding site of small-molecule inhibitors[19]. The cysteine residue Cys522 only occurs in FGFR4 but not in other FGFR members. Targeting the unique cysteine 552, located in the hinge region of FGFR4, provides a selectivity handle to specifically target FGFR4 without affecting FGFR1-3. Selective FGFR4 inhibitors, such as BLU9931, BLU554, and FGF401, have been developed to target the unique cysteine 552 of FGFR4[19–21]. Recently, the BLU554 (fisogatinib) phase I trial showed that gatekeeper (Val550) and hinge-1 (Cys552) residues in the kinase domain of FGFR4 were identified mutations in two HCC patients treated with BLU554[22]. These mutations lead to drug resistance by forming conflicts and preventing covalent binding between FGFR4 and BLU554[22]. This phenomenon also suggests that new drug-resistant mutations may lead to ineffective FGFR4-specific inhibitors against Cys552[22]. In addition to potential drug resistance, there are six kinases that contain the cysteine equivalent to Cys552 of FGFR4, which might cause off-target effects (Fig. 1b). Therefore, a novel and unique binding mode of FGFR4 inhibitors are needed to overcome the target resistance mutations and further improve its selectivity for FGFR4.

A number of FGFR/inhibitor complex structures have been determined[2]. Based on the structural observation and sequence analysis, two cysteines (Cys477 and Cys552) in the FGFR4 kinase domain make it possible to design a dual-warhead covalent FGFR4 inhibitor. In this study, we designed and synthesized the first dual-warhead covalent FGFR4 inhibitor (CXF-009, Fig. 1c) targeting two cysteines in the FGFR4 ATP-binding pocket. Our study will provide a novel research direction for the next generation of FGFR4 inhibitors to treat HCC or other FGFR4-dependent cancers.

## Results

**Chemistry.** To obtain a dual-warhead covalent FGFR4 inhibitor, **CXF-009** maintaining the pyrimido[4,5-d] pyrimidinone core scaffold was designed and optimized based on the structure of the covalent FGFR inhibitor FIIN-2[13]. **CXF-008** and **CXF-009**

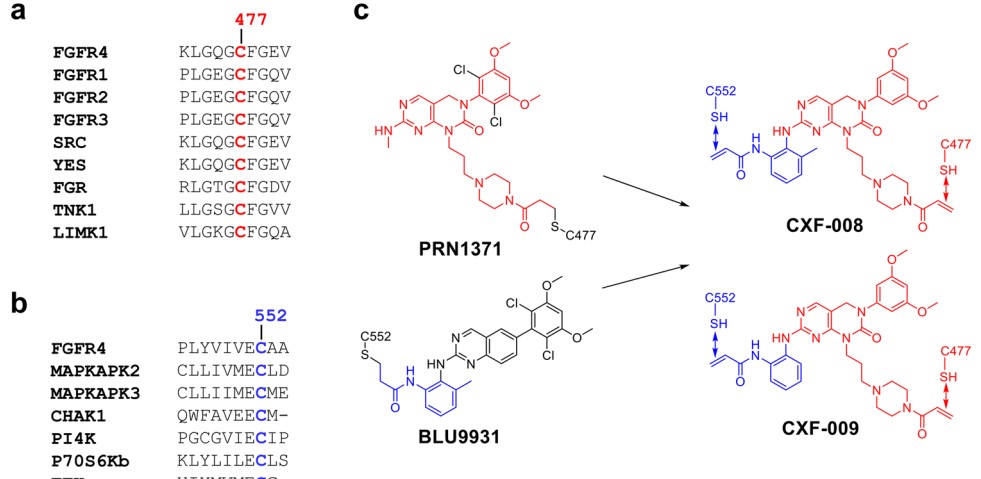

**Fig. 1 Design of dual-warhead covalent inhibitor of FGFR4. a, b** Kinases with the cysteines at the equivalent position of Cys477 (**a**) and Cys552 (**b**) in FGFR4 within the human kinome. **c** Design the new dual-warhead covalent inhibitors **CXF-009** and **CXF-008** for Cys477 and Cys552 of FGFR4. 4-acrylamidepiperazin-1-propyl from PRN1371 is colored red, and the 2-acrylamidebenzamine group from BLU9931 is indicated in blue.

**Fig. 2 Synthesis of compound CXF-008 and CXF-009. CXF-008** and **CXF-009** were synthesized using 5-hydroxymethyluracil as the starting material via eight steps.

possess two arms that can both reach cysteine residues located in the P-loop and hinge regions of FGFR4. One arm is 4-acryla-midepiperazin-1-propyl, which is from PRN1371 and could directly form covalent bonds with the P-loop Cys477 of FGFR4. The other is the 2-acrylamidebenzamine group from BLU9931, which binds the hinge Cys552 of FGFR4 with a covalent bond (Fig. 1c). **CXF-008** and **CXF-009** were synthesized using 5-hydroxymethyluracil as the starting material in eight steps (Fig. 2 and Supplementary Figs. 1, 2 for NMR spectra).

**CXF-009 is a dual-warhead covalent inhibitor of FGFR4.** To verify the dual-warhead covalent bonds within FGFR4 and the two inhibitors, mass spectrometry was first performed. The molecular weight of FGFR1 was 35215 Da, and the molecular weight of the FGFR1-**CXF-009** complex was 35842 Da, which indicated that **CXF-009** formed a covalent bond with FGFR1 (Supplementary Fig. 3). **CXF-008** also forms a covalent bond with FGFR1, and the molecular weight of the FGFR1-**CXF-008** complex is 35841 Da (Supplementary Fig. 3). As shown in Fig. 3a, the molecular weight of FGFR4 was 34652 Da, and a mass shift was displayed after **CXF-009** incubation. The spectrum confirmed the covalent binding of FGFR4-CXF-009 with an m/z centered at 35279 Da, consistent with the sum of the molecular weights of the inhibitor and kinase. To determine which cysteines of FGFR4 form covalent bonds with compound **CXF-009**, we mutated the cysteines (Cys477 and Cys552) in the ATP-binding pocket of FGFR4. Based on the mass shift in the mass spectrometry, **CXF-009** forms covalent bonds with FGFR4(C477A) and FGFR4(C552A), respectively (Fig. 3b, c). No mass shift was observed in the mass spectrometry after FGFR4(C477A, C552A) incubation with **CXF-009** (Fig. 3d). **CXF-008** forms a covalent bond with FGFR4(C477A) but not with FGFR4(C552A), which means that **CXF-008** only forms one covalent bond with Cys552 of FGFR4 (Supplementary Fig. 4). Overall, these results suggest that **CXF-009** is a dual-warhead covalent inhibitor of FGFR4.

**Overall structure of CXF-009 in complex with FGFR4.** To further understand the binding mode and structure-activity relationship between **CXF-009** and FGFR4, we solved the co-

crystal structure of the FGFR4 kinase domain bound to **CXF-009** at 2.0 Å resolution (PDB ID: 7V29). **CXF-009** binds within the ATP-binding pocket of FGFR4 and forms covalent bonds with the Cys477 residue located within the p-loop and the Cys552 residue located within the hinge loop, respectively (Fig. 4a). The electron density of **CXF-009** is well-defined in the crystal structure (Fig. 4b). In addition to these two covalent bonds, **CXF-009** also forms multiple hydrogen bonds with Arg483 and Ala553 in the hinge loop of FGFR4 (Fig. 4c). The amethoxy group of **CXF-009** forms a hydrogen bond with the main chain NH group of Asp630 (Fig. 4c). van der Waals contacts are also formed to further enhance the interactions between **CXF-009** and FGFR4.

**CXF-009 is a potent and specific FGFR4 inhibitor.** To test the inhibitory selectivity and efficiency of **CXF-009** against FGFR4, we first performed an in vitro kinase inhibitory activity assay on four protein members of the FGFR family. As shown in Fig. 5a, **CXF-009** demonstrated robust potency against FGFR4 with an $IC_{50}$ of 48 nM, whereas the $IC_{50}$ values of **CXF-009** to FGFR1-3 were 2579 nM, 2408 nM, and 977 nM, respectively, at least 20-fold higher than that of FGFR4. The comparison with the positive control, BLU9931, and PRN1371, against FGFR4 was shown in Supplementary Fig. 5. The $IC_{50}$ values of **CXF-009** against FGFR4(C477A), FGFR4(C552A), and FGFR4(C477A, C552A) were 193 nM, 602 nM, and 1528 nM, respectively, which are 4-fold, 12-fold, and 31-fold lower than the activity against the wild type, respectively (Fig. 5a). These results further supported that a dual-warhead covalent bond was formed within the **CXF-009** and FGFR4, and **CXF-009** had a stronger inhibition than FGFR4(C552A).

Then, we further measured the potency of **CXF-009** to inhibit the proliferation of Ba/F3 cells engineered to be dependent on FGFR1-4 activity. **CXF-009** showed negligible inhibitory activity on parental Ba/F3 cells with $IC_{50}$ values above 5000 nM. **CXF-009** inhibited the growth of Ba/F3 cells transformed with FGFR1-4 with $IC_{50}$ values of 1562 nM, 1971 nM, 777 nM, and 38 nM, respectively, which is consistent with the results of the in vitro kinase assay (Fig. 5b). We also measured the inhibitory potency of **CXF-009** on two FGF19-overexpressing HCC cell lines, Hep3B and Huh7. The $IC_{50}$ values of **CXF-009** on these two HCC cells lines were 895 nM and 727 nM, respectively (Supplementary

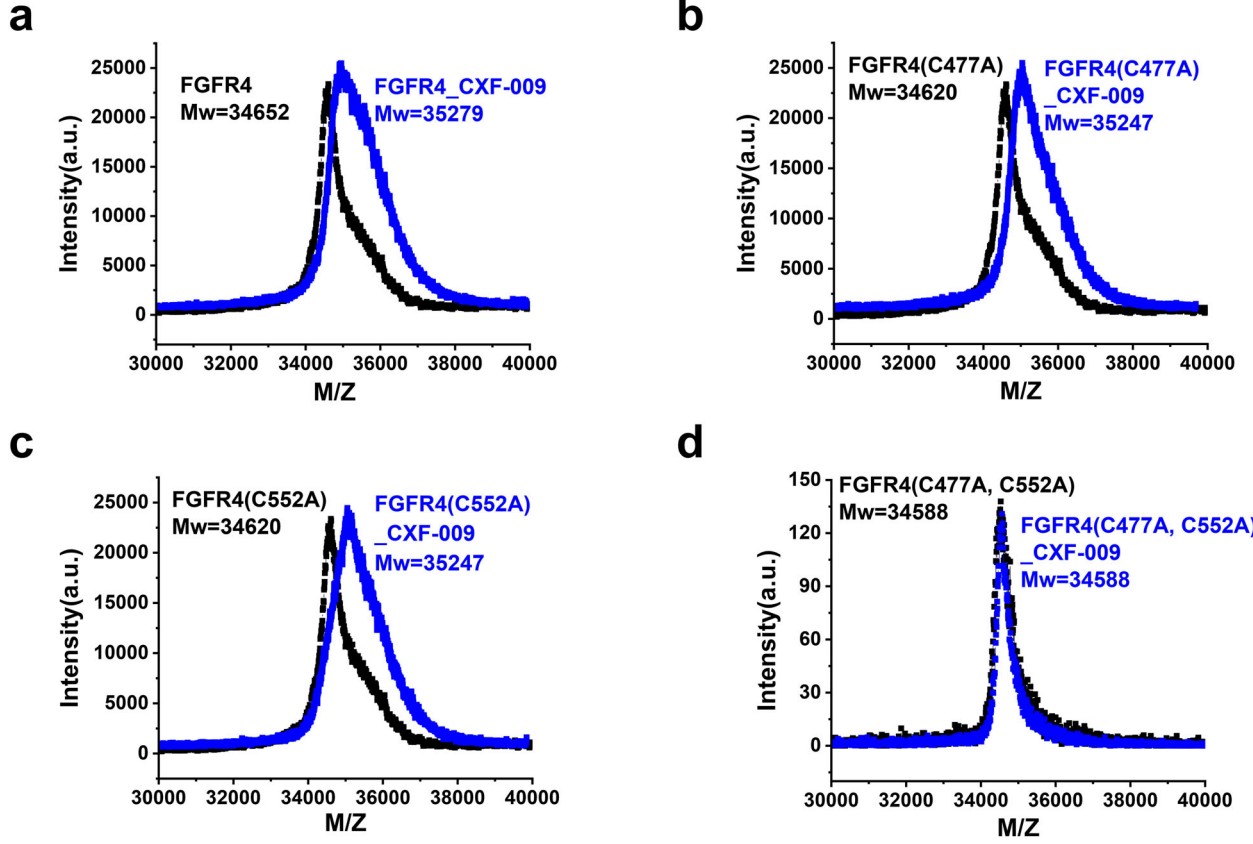

**Fig. 3 MALDI-TOF MS determination of protein (black) and protein/inhibitor complexes (blue) for CXF-009. a** Apo FGFR4 and FGFR4/**CXF-009** mixture; **b** Apo FGFR4(C477A) and FGFR4(C477A)/**CXF-009** mixture; **c** Apo FGFR4(C552A) and FGFR4(C552A)/**CXF-009** mixture; **d** Apo FGFR4(C477A, C552A) and FGFR4(C477A, C552A)/**CXF-009** mixture.

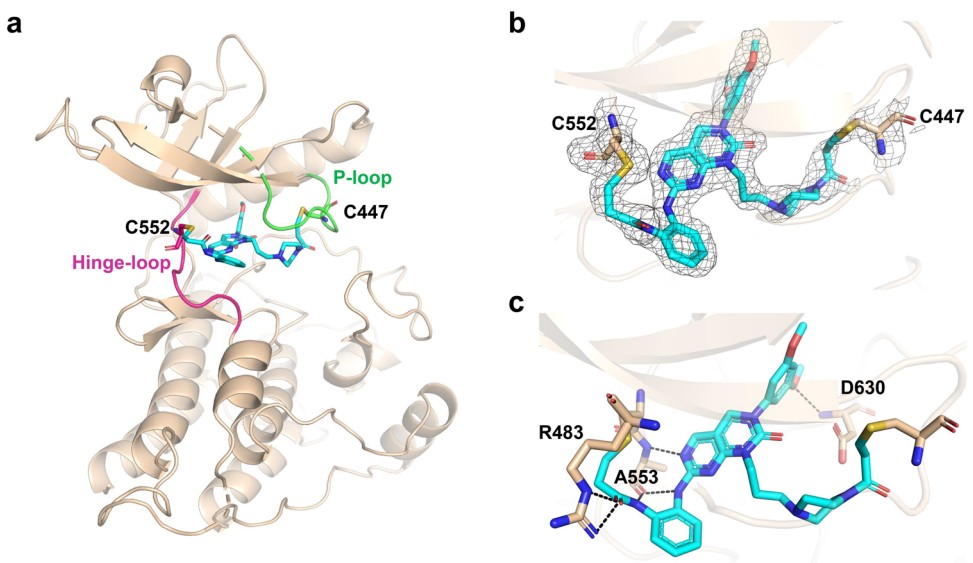

**Fig. 4 Crystal structure of FGFR4/CXF-009 complex. a** The overall structure of FGFR4/**CXF-009** complex. FGFR4 kinase is colored as wheat, and **CXF-009** is shown as cyan. Hinge loop (warm-pink) and P-loop (green) are high-light in the structure. **b** The 2Fo-Fc omit map (black mesh, contoured at 0.7σ) of the FGFR4/**CXF-009** co-crystal structure. **c** The hydrogen bonds between FGFR4 and **CXF-009**. Hydrogen bond distance is <3.3 Å.

Fig. 6). Altogether, these results show that **CXF-009** is a potent and paralog-selective inhibitor of FGFR4.

**CXF-009 is a highly selective FGFR4 inhibitor**. To determine the inhibitor's kinome wide selectivity, **CXF-009** was profiled at

1 μM for inhibition against a panel of 185 human kinases using a microfluidic screening platform. This analysis revealed that **CXF-009** caused significant inhibition of only FGFR4 out of the 185 kinases (Fig. 5c). The IC$_{50}$ value of **CXF-009** against FGFR4 is 74.51 nM by HTRF assay (Supplementary Fig. 7). Moreover, the difference between inhibition against FGFR4 of **CXF-009** (94%)

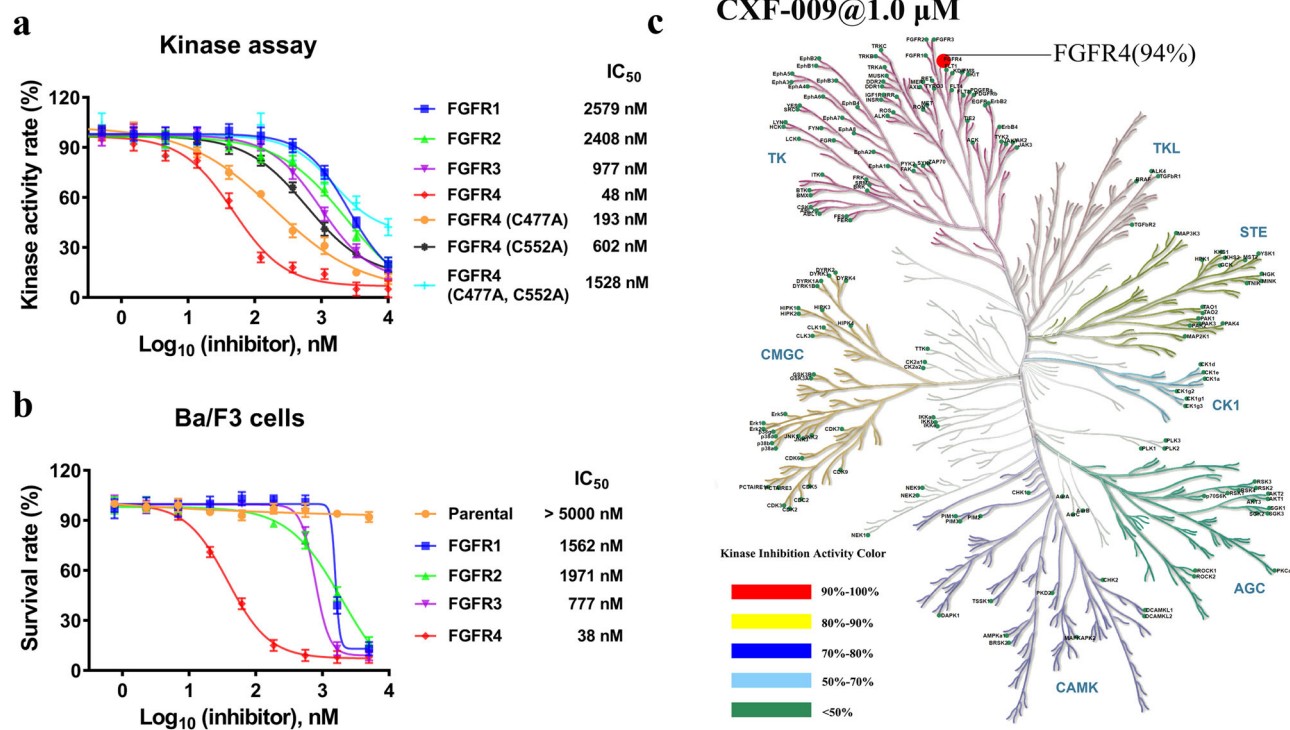

**Fig. 5 CXF-009 selectively inhibited FGFR4. a** Potency and selectivity of CXF-009 against wild-type FGFR1-4 and FGFR4 mutations using kinase assay. **b** Cellular activity determination of **CXF-009** using a Ba/F3 cell model. Data are presented as dot, $n = 3$ independent replicates. Error bars indicate mean ± SD. **c** Kinase inhibition profile of **CXF-009** with 185 kinases at 1000 nM.

| Table 1 The in vitro pharmacokinetic parameters of CXF-009. | | |
|---|---|---|
| | **Parameters** | **Value** |
| Rat plasma | Regression equation | $y = -0.1388x + 4.604$ |
| | R squared | 0.9935 |
| | Slope | −0.1388 |
| | $t_{1/2}$ | 4.99 h |
| Human plasma | Regression equation | $y = -0.1859x + 4.622$ |
| | R squared | 0.9969 |
| | Slope | −0.1859 |
| | $t_{1/2}$ | 3.73 h |

and that of the other 184 kinases (<50%) was rather large (Supplementary Table 2). These results confirm that **CXF-009** is a potent, specific, and highly selective FGFR4 inhibitor.

**In vitro pharmacokinetic assay.** We first calculated the in vitro pharmacokinetic parameters (in vitro $t_{1/2}$) of **CXF-009** using in vitro pharmacokinetic assay, by the quantification of the **CXF-009** concentration after incubation with rat/human plasma for specific time intervals. The **CXF-009** plasma stability curve was depicted by plotting the percentage remaining **CXF-009** against incubation time (Supplementary Fig. 8A, C). From the constructed curve, the percentages that exhibited linearity (0–10 h) were selected to plot another curve of the natural logarithm (ln) of the percentage remaining **CXF-009** against incubation time (Table 1, Supplementary Fig. 8B, D). Based on equation[23], the **CXF-009** in vitro $t_{1/2}$ was 4.99 h in rat plasma and 3.73 h in human plasma (Table 1), exhibiting a medium excretion ratio. These results indicated that **CXF-009** has potentially good bioavailability and a moderate possibility of drug accumulation in the body.

**CXF-009 is slightly more reactive to GSH than PRN1371**. As a dual-warhead covalent inhibitor of FGFR4, we performed a GSH reaction assay to assess whether the two warheads of **CXF-009** increase its toxicity. The GSH reaction assay was performed to evaluate the reactive rate between **CXF-009** and GSH. We measured the remaining GSH in the luminescent reaction scheme with the DTNB probe. The results showed that **CXF-009** and PRN1371 induced a time-dependent decrease in GSH with similar trends and CXF-009 is slightly more reactive to GSH than PRN1371 (Supplementary Fig. 9).

**Discussion**

FGFR4 aberration has been shown to contribute to carcinogenesis, metastasis, and resistance to anticancer agents, making FGFR4 a promising target to treat FGF19/FGFR4-dependent cancers[15]. To date, pan-FGFR inhibitors lack sufficient selectivity, limiting their feasible use due to serious side effects associated with inhibiting activity against other kinases[21]. In addition, selective FGFR4 inhibitors targeting Cys552 induced new resistance mutations, leading to ineffective FGFR4-specific inhibitors in a phase I trial[20]. Referring to the development process of EGFR inhibitors[24], the development of FGFR inhibitors may have to undergo targeting amino acid mutations as well. The drug resistance of selective FGFR4 inhibitors predicts the urgent need to develop novel binding modes of selective FGFR4 inhibitors[18].

In this study, we designed and synthesized a dual-warhead covalent inhibitor, **CXF-009**, of FGFR4 that covalently targets Cys477 and Cys552 residues specific to FGFR4 (Figs. 1 and 2). Our mass spectrometry results and structural analysis both show that **CXF-009** forms dual-warhead covalent bonds with FGFR4 (Fig. 3). Dual-warhead covalent bonds were observed between the compound **CXF-009** and the Cys477 and Cys552 residues of FGFR4 (Fig. 4). To our knowledge, **CXF-009** is the first reported dual-warhead covalent inhibitor targeting FGFR.

As **CXF-009** covalently binds to the Cys477 and Cys552 of FGFR4 at the same time, **CXF-009** may possess highly favorable potency and selectivity. **CXF-009** exhibited exquisite selectivity toward FGFR4 within the FGFR family by showing greater inhibition of this paralog over FGFR1-3 (Fig. 5a, b). In addition, **CXF-009** exhibited exquisite selectivity for FGFR4 within the human kinome (Fig. 5c). These results suggest that **CXF-009** shows strong selectivity against FGFR4 (Fig. 5).

The dual-warhead covalent binding of **CXF-009** to FGFR4 was also indirectly confirmed in vitro by exhibiting a fourfold greater inhibition of WT FGFR4 kinase over a cysteine mutant (C477A) and 13-fold greater inhibition over the C552A mutant (Fig. 5a). In contrast, those inhibitors targeting Cys552, such as FGF401 and BLU554 exhibited much less-potent inhibition (IC$_{50}$ > 10 $\mu$M) of the FGFR4(C552A) mutant[21,25]. This means that CXF-009 could overcome the drug resistance induced by a single cysteine mutant. When two cysteines (Cys477 and Cys552) were mutated, **CXF-009** almost lost its inhibition of FGFR (IC$_{50}$ > 1 $\mu$M, Fig. 5A).

As we know, although covalent drugs have advantages in drug resistance still need to face the problem of drug toxicity brought by the covalent warhead. As **CXF-009** has two warheads, whether the toxicity will be increased needs to be evaluated. We compared the GSH reactive rate of **CXF-009** with that of PRN1371 using a GSH reaction assay. **CXF-009** and PRN1371-depleted GSH with similar trends and **CXF-009** are slightly more reactive to GSH (Supplementary Fig. 9). In addition, in our Ba/F3 experiments, **CXF-009** is potent against FGFR4-overexpressing Ba/F3 cell (IC$_{50}$ = 38 nM), whereas it has no effect on parental Ba/F3 cells with IC$_{50}$ > 5 $\mu$M (Fig. 5b). These data suggest that CXF-009 may provide good antitumor efficacy under a safety window.

In summary, **CXF-009** represents a dual-warhead covalent FGFR4 inhibitor that might act as an effective therapy for a subset of patients driven by aberrant FGFR4 signaling and overcome the drug-induced resistance mutations. Our findings suggest that the dual-warhead covalent FGFR4-specific inhibitor described here may serve as a promising lead compound for future anticancer drug discovery.

## Methods

**Synthetic procedures**. See Supplementary Methods.

**Protein expression and purification**. Wild-type FGFRs and mutant FGFRs were prepared as previously described[25–27]. In brief, the kinase domains of human FGFR1 (residues 456–765), FGFR2 (residues 453–770), FGFR3 (residues 450–758), and FGFR4 (residues 445–753) were cloned into a modified pET28a expression vector and then transfected into *Escherichia coli* Rosseta cells. The N-terminal of the recombinant proteins were contained a 6xHis tag followed by a PreScission cleavage site for FGFR1, FGFR2, FGFR3, and FGFR4. Site-directed mutagenesis of FGFR4 was generated according to the ClonExpress II One Step Cloning Kit (Vazyme) using the pET28a-FGFR4 plasmid as the template[26]. These mutants were confirmed by DNA sequencing (Tsingke, Changsha, China). For crystallization experiments, the wild-type FGFR4 plasmid was co-expressed with untagged YOPH to obtain non-phosphorylated proteins. The proteins were first purified by Ni-NTA affinity chromatography. The 6 × His tag was removed by PreScission protease. Then, anion exchange chromatography (Mono Q) and size exclusion chromatography (if necessary) were exploited for further purification. The target proteins were concentrated to ~5–15 mg/mL and flash frozen for storage at −80°C for subsequent studies.

**MALDI-TOF-MS**. Matrix-assisted laser desorption/ionization time-of-flight mass spectrometry (MALDI-TOF-MS) was performed to determine the molecular weight of the samples. Sinapic acid (Sigma Part no. #85429) was used as the matrix on the instrument (AB SCIEX 5800). The operation mode was reflective mode with positive ion detection. The delayed extraction time was 200 ns, whereas the accelerating voltage was 19,000 V. Desalting samples (<1 mg/mL) were mixed with sinapic acid (20 mg/ml) at a volume ratio of 1:1. The mass spectrogram was generated using Data Explorer and Origin.

**Crystallization**. Crystals were obtained by the hanging-drop vapor diffusion method. Protein-inhibitor complexes were prepared by mixing protein with inhibitor at a 1:1.5 molar ratio. Crystals of FGFR4/**CXF-009** were grown under 0.1 M Bis-Tris (pH 4.5), 0.2 M Li$_2$SO$_4$, 18% PEG3350 at 4°C for a week. Prior to diffraction experiments, the crystal was cryoprotected by supplementing the mother liquor with 20% glycerol, then cooled in liquid nitrogen.

**Data collection and structure determination**. All data sets were collected at 100 K at the beamline BL17U1 of Shanghai Synchrotron Radiation Facility (SSRF) (wavelength :0.97915)[28]. The diffraction data were processed using the HKL3000[29]. The structure was solved by molecular replacement using PHENIX Phaser[30] with the FGFR4/ponatinib (PDB ID: 4QRC[31]) used as an initial search model. The structure was refined by iterative reciprocal and real space refinement using PHENIX, and at least one round of cartesian simulated annealing refinement in PHENIX prior modeling ligands to avoid phase bias. Further refinements were done by Phenix and CCP4[30,32]. The statistics of the data collection and structural refinements are presented in Supplementary Table 1. Graphical representations of structures were generated using PyMol[33].

**Kinase inhibition assay**. All kinase inhibition assays were performed with optimized kinase assay buffer consisting of 40 mM Tris-HCl pH 7.5, 20 mM MgCl$_2$, 20 mM NaCl, 0.1 mg/mL BSA, 1 mM TCEP, and 4% DMSO. Diluted concentrations (ranging from 25 $\mu$M to 0.01 nM) of inhibitor were incubated with kinases (0.1 $\mu$M) in 384-well plates. In all, 5 × ATP plus poly (4:1 Glu, Tyr) peptides were added to start the kinase reactions. Reactions were terminated by the addition of stop buffer ADP-Glo after 30 min of incubation. Fluorescence was measured on a multimode plate reader (Perkin-Elmer) using a chemiluminescence program after the addition of the detection reagent. IC$_{50}$ values were determined using a three-parameter log [Inhibitor] versus response model in GraphPad Prism software.

**Ba/F3 cell viability assay**. The TEL-FGFRs gene, constructed by fusing the kinase domain of FGFRs in frame with the extracellular domain of TEL, was cloned into a modified pMSCV-IresGFP expression vector and then transfected into Ba/F3 cells (Cell Resource Center, IBMS, CAMS, China)[34]. The transformed Ba/F3 cells were seeded in a 96-well plate and treated with each concentration of the compounds. After 72 h the cells were assessed by a CCK-8 kit (Vazyme, China). The IC$_{50}$ values were calculated using GraphPad Prism software.

**Hep3B and Huh7 viabilities assay**. Hep3B (ThermoFisher Scientific, 11965, America) and Huh7 (ThermoFisher Scientific, 11095, America) cells were seeded into 96-well plates at 4000 cells/well and 2000 cells/well, respectively. Cells were treated with triple dilution **CXF-009** for 72 h. Cell viabilities were measured by a CCK-8 kit (Vazyme, China). The IC$_{50}$ values were calculated using GraphPad Prism software.

**Kinase selectivity profiling**. CXF-009 was screened at a single concentration of 1 $\mu$M against a panel of 185 diverse protein kinases (Supplementary Table 2) using the microfluidic screening platform at ICE Bioscience.

**In vitro plasma stability study**. CXF-009 was spiked with rat/human plasma to obtain a final concentration of 2 $\mu$g/mL and then suffered from 37°C incubation for 0, 0.5, 1, 2, 4, 6, 8, 10, 24, 36, and 48 h. The incubation was terminated by the addition of 300 $\mu$L of acetonitrile containing 50 ng/mL roblitinib as internal standard. After vortex and centrifugation, the supernatant was analyzed using ultra-performance liquid chromatography-tandem mass spectrometry.

**GSH reaction assay**. GSH (5 mM) and inhibitor (1 mM) or DMSO control were mixed in equal volumes and reacted at 37°C for 10 min, 20 min, 30 min, 40 min, 50 min, and 60 min. The remaining GSH was detected with DTNB (Beyotime, China), and the absorption light was measured at 412 nm on a plate reader (Perkin-Elmer). Data were determined using the time (min) versus % of control in Origin 2018 software.

**Reporting summary**. Further information on research design is available in the Nature Research Reporting Summary linked to this article.

## Data availability

The coordinates and structural factors have been submitted to the Protein Data Bank with accession codes 7V29. The authors declare that all data supporting the findings of this study are available within the paper and its Supplementary Information files. All raw data are available from the corresponding author upon request.

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

## Acknowledgements

This work was supported by the National Natural Science Foundation of China (grants 81570537, 81974074, and 82172654 to Y.C.; grant 31900880 to H.W.), China Post-doctoral Science Foundation (2019M652805), Science and Technology Planning Project of Hunan Province (2018TP1017), National Science Foundation of Hunan Province (grants 2021JJ40961). We thank the staff from the BL17U/BL19U1 beamline of the National Facility for Protein Science in Shanghai (NFPS) at SSRF for assistance during data collection.

## Author contributions

X.C. performed a biological experiment; H.L. synthesized compound; M.L. performed in vitro plasma stability study; Q.L., S.D., S.Y., and L.Q. performed structure determination. X.C., M.G., H.W., and J.L. analyzed the data. X.C., L.J., G.X., and Y.C. prepared and revised the manuscript.

## Competing interests

The authors declare no competing interests.
