## [Peer Review File · Communications Chemistry]

Reviewers' comments:

Reviewer #1 (Remarks to the Author):

Comments

Chen and co-authors discovered a novel covalent inhibitor of FGFR4. X-ray structure shows the compound can covalently bind to two Cys residues in active site. Moreover, the compound has high selectivity according to the kinase panel data. The results are very interesting. I would like to recommend to publish it after some revisions and the following questions are addressed.

(1) the compound structure indeed is a combination of two reported FGFR4 inhibitors and only one compound was stated in the paper. Author should synthesized more compounds for screening and SAR studies.

(2) What's the advantage for formation of two covalent bonds? For FGFR4 kinase assays, the comparison with positive control was needed (PRN1371 and BLU9931).

(3) there were two Michael-receptors in the compound, I am afraid it is more reactive to GSH. Authors should evaluate the reactive rate between compound and GSH. Additionally, PK data should also be provided.

Reviewer #2 (Remarks to the Author):

In this work, the authors provide a novel dual-warhead covalent FGFR4 inhibitor CXF-009 that can covalently target two cysteines Cys477 and Cys552 of FGFR4. They believe CXF-009 is the first reported inhibitor that forms dual-warhead covalent bonds with two cysteine residues in FGFR4 and has the potential to overcome drug induced resistant FGFR4 mutations. The discovery may of some interest to Communications Chemistry. However, there are some issues need to be further addressed.

1) As we know, although covalent drugs has its advantages in drug resistance but still need to face the problem of drug toxicity brought by the covalent warhead. As CXF-009 has two warheads, whether the toxicity will be increased and how to promise the antitumor efficacy under a safety window need to be evaluated and discussed in detail.

2) As the author mentioned, FGFR4 is related to HCC, but they only measured the potency of CXF-009 to inhibit the proliferation of Ba/F3 cells engineered to be dependent on FGFR1-4 activity. They'd better evaluate the inhibitory potency of CXF-009 on HCC specific cells such a Huh7, Hep3B, HepG2 etc.

3) For Extended Data Fig. 2. The IC₅₀ of CXF-009 against FGFR4 using HTRF assay. Only two points have error bars. How many duplicate experiments were tested should be mentioned. And error bars for all points should be included.

4) In many parts of the manuscript, "IC₅₀ value" was used. It should be changed to "IC₅₀ value"

Dear Dr. Bissette,

We are grateful to you and the reviewers for providing constructive comments on our manuscript. We are pleased to submit a revised version of this manuscript with modifications, as suggested by you and the reviewers.

Response to Reviewers' Comments

Reviewer #1:

(1) the compound structure indeed is a combination of two reported FGFR4 inhibitors and only one compound was stated in the paper. Author should synthesized more compounds for screening and SAR studies.

Response: Thank you for your suggestion. In our study, we also synthesized and tested another compound **CXF-008** (Figure 1C). Although **CXF-008** has two warheads, our study shows that **CXF-008** can only form one covalent bond with Cys552 of FGFR4, but not with Cys477 of FGFR4 (Supplementary Fig. 1 and Supplementary Fig. 2).

(2) What's the advantage for formation of two covalent bonds? For FGFR4 kinase assays, the comparison with positive control was needed (PRN1371 and BLU9931).

Response: The advantage for the formation of two covalent bonds are as follows:

1) Two covalent bonds formed between **CXF-009** and FGFR4 may increase the selectivity of **CXF-009**. Cys477 of FGFR4 is only conserved in 8 other kinases (including FGFR1-3) within the human kinome (Figure 1A). Cys552 of FGFR4 is only conserved in 6 other kinases within the human kinome (Figure 1B). Only FGFR4 has both cysteines, therefore formation of two covalent bonds may provide the ultimate selectivity of FGFR4.

2) Formation of two covalent bonds could overcome the drug resistance induced by a single cysteine mutation. If one cysteine is mutated, the compound could still be effective since it could still form a covalent bond with the other cysteine.

In the revised manuscript, we have added positive control for FGFR4 kinase assays (Supplementary Figure 3)

(3) there were two Michael-receptors in the compound, I am afraid it is more reactive to GSH. Authors should evaluate the reactive rate between compound and GSH. Additionally, PK data should also be provided.

Response: Thank you for the suggestion. In the revised manuscript, we have evaluated the reactive rate between **CXF-009** and GSH (Supplementary Fig. 7). Compared to PRN1371, **CXF-009** is slightly more reactive to GSH. In addition, we have provided in vitro PK data (Table1 and Supplementary Fig. 6).

Reviewer #2 (Remarks to the Author):

1) As we know, although covalent drugs has its advantages in drug resistance but still need to face the problem of drug toxicity brought by the covalent warhead. As **CXF-009** has two warheads, whether the toxicity will be increased and how to promise the antitumor efficacy under a safety window need to be evaluated and discussed in detail.

Response: Thank you for the comment. In the revised manuscript, we compared the GSH reactive

rate of **CXF-009** with that of PRN1371 using a GSH reaction assay. **CXF-009** and PRN1371 depleted GSH with similar trends and **CXF-009** is slightly more reactive to GSH (Supplementary Fig. 7). In addition, in our Ba/F3 experiments, **CXF-009** is potent against FGFR4-transformed Ba/F3 cells ($IC_{50}=38$ nM), while it has no effect on parental Ba/F3 cells with $IC_{50}>5$ μ M. These data suggest that **CXF-009** may provide good antitumor efficacy under a safety window. Further studies are needed to test its toxicity in vivo. We have added this part in the Discussion section in the revised manuscript.

2) As the author mentioned, FGFR4 is related to HCC, but they only measured the potency of CXF-009 to inhibit the proliferation of Ba/F3 cells engineered to be dependent on FGFR1-4 activity. They'd better evaluate the inhibitory potency of CXF-009 on HCC specific cells such as Huh7, Hep3B, HepG2 etc.

Response: Thank you for the comment. In the revised manuscript, we have evaluated inhibitory potency of **CXF-009** on HCC specific cells such as Huh7 and Hep3B (Supplementary Fig. 4). The IC_{50} values of **CXF-009** on Hep3B and Huh7 cell lines were 895 nM and 727 nM, respectively.

3) For Extended Data Fig. 2. The IC_{50} of CXF-009 against FGFR4 using HTRF assay. Only two points have error bars. How many duplicate experiments were tested should be mentioned. And error bars for all points should be included.

Response: Thank you for the comment. In Extended Data Fig. 2, we indeed included error bars for all points. However only two points are obvious, because the error bars for other points are very small. To address the reviewer's comment, we have remade and enlarged the figure (Supplementary Fig. 5). Three independent experiments were tested. We have added this information in the revised manuscript.

4) In many parts of the manuscript, "IC50 value" was used. It should be changed to "IC₅₀ value"

Response: Thank you for your suggestion. We have changed all the "IC50 value" to "IC₅₀ value".

We thank you and the reviewers again for your considerable efforts in reviewing our manuscript.

Sincerely,

Yongheng Chen, Ph.D.

REVIEWERS' COMMENTS:

Reviewer #1 (Remarks to the Author):

Based on the feedbacks and revisions from Authors, I would like to recommend the article to be accepted for publication in Communications Chemistry. However, The toxicity of covalent inhibitor is often problematic, It is strongly suggested that the authors can do the animal safety test in future study.

Reviewer #2 (Remarks to the Author):

Most of the points have been addressed and now can be published.
But "IC50 value" hasn't been revised all and should be revised.

Dear Dr. Victoria Richards,

We are grateful to you and the reviewers for providing constructive comments on our manuscript. We are pleased to submit a revised version of this manuscript with modifications, as suggested by you and the reviewers. We also edited our revised manuscript to comply with your journal policies and formatting style.

Response to Reviewers' Comments

Reviewer #2 (Remarks to the Author):

But "IC50 value" hasn't been revised all and should be revised.

Response: Thank you for the suggestion. All the "IC50 value" have been revised to "IC₅₀ value" in the revised manuscript.

We thank you and the reviewers again for your considerable efforts in reviewing our manuscript.

Sincerely,

Yongheng Chen, Ph.D.